# An Annotated Checklist of Invasive Species of the Phyla Arthropods and Chordates in Panama

**DOI:** 10.3390/biology13080571

**Published:** 2024-07-28

**Authors:** Digna Rodríguez-Gavilanes, Humberto A. Garcés Botacio, Rogemif Fuentes, Louise Rodriguez-Scott, Yostin Añino, Oscar G. López-Chong, Enrique Medianero

**Affiliations:** 1Escuela de Biología, Universidad de Panamá, Campus Octavio Méndez Pereira, Panama 0819-07289, Panama; dignamr09@gmail.com; 2Departamento de Biología Marina y Limnología, Universidad de Panamá, Campus Octavio Méndez Pereira, Avenida Simón Bolívar, Panama 0819-07289, Panama; humberto.garces@up.ac.pa; 3Fundación Los Naturalistas, David, Chiriquí 0426-01459, Panama; rogemifdaniel@gmail.com; 4Programa de Maestría en Ciencias Biológicas, Facultad de Ciencias, Naturales, Exactas y Tecnología, Universidad de Panamá, Panama 0819-07289, Panama; louise.rodriguez001@outlook.com; 5Museo de Invertebrado G.B.Fairchild, Universidad de Panamá, Panama 0819-07289, Panama; yostin0660@gmail.com; 6Colección de Aves, Instituto Smithsonian de Investigaciones Tropicales, Panama 0843-03092, Panama; lopezog@si.edu; 7Departamento de Ciencias Ambientales y Programa de Maestría en Entomología, Universidad de Panamá, Campus Octavio Méndez Pereira, Avenida Simón Bolívar, Panama 0819-07289, Panama; 8Miembro del Sistema Nacional de Investigación (SNI-SENACYT), Panama 0816-02852, Panama

**Keywords:** invasive species, introduced species, biodiversity, Central America, tropical forest, pest

## Abstract

**Simple Summary:**

Invasive species are considered a threat to the conservation of different environments. Annotating the numbers and species of these invasive organisms is critical to developing conservation strategies. This research gives background information on the types and possible origins of invasive species from the arthropod and chordate groups in Panama. The results indicated that approximately 141 exotic arthropod and chordate species have been reported as invasive species in Panama. Most of these species are believed to have been introduced via the Panama Canal Zone or accidentally. With the information compiled, this study will serve as preliminary data on the sources of introduction and will provide information for future research and plans to prevent the impact of those species.

**Abstract:**

Invasive species are one of the five main causes of biodiversity loss, along with habitat destruction, overexploitation, pollution, and climate change. Numbers and species of invasive organisms represent one of the first barriers to overcome in ecological conservation programs since they are difficult to control and eradicate. Due to the lack of records of invasive exotic species in Panama, this study was necessary for identifying and registering the documented groups of invasive species of the Chordates and Arthropod groups in Panama. This exhaustive search for invasive species was carried out in different bibliographic databases, electronic portals, and scientific journals which addressed the topic at a global level. The results show that approximately 141 invasive exotic species of the Arthropoda and Chordata phyla have been reported in Panama. Of the 141 species, 50 species belonged to the Arthropoda phylum and 91 species belonged to the Chordate phylum. Panamanian economic activity could facilitate the introduction of alien species into the country. This study provides the first list of invasive exotic chordate and arthropod species reported for the Republic of Panama.

## 1. Introduction

Invasive species are one of the five main causes of biodiversity loss, along with habitat destruction, overexploitation, pollution, and climate change [1,2]. An invasive species is understood to be an exotic species that establishes itself in a natural or semi-natural ecosystem or habitat and is an agent inducing changes that affect native biodiversity [1]. The process of invasion is a progressive phenomenon that does not have to be unidirectional [3]. This means that not all introduced species (exotic) will become naturalized, nor will all naturalized ones become invasive. Additionally, there is not a consistent number of species that goes from one phase to another [3].

The problems associated with invasive species are summarized in three main aspects [2]: (1) big economic losses and ecological impacts caused by many invasive species; (2) the increase in the number of introduced species that become invasive and the problems associated with these species; and (3) the need to include invasive species in any ecological study, keeping in mind that these organisms are able to alter an ecosystem in which they are introduced [2].

Invasive species are one of the first challenges to overcome in ecological conservation programs since they are difficult to control and eradicate, mainly due to their reproductive strategies combined with their dispersal, establishment, and persistence [2]. Invasive alien species can be found in all groups of organisms [4]. Globally, 90% of vertebrate and plant introductions are intentional, and the remaining 10% are accidental [5]. After plants, arthropods and chordates include the largest number of invasive exotic species reported for Latin America and the Caribbean [4]. The animal groups which pertain to the Phylum Arthropoda contain the largest number of species known today [6]. Some species transmit diseases or are vectors of them, and some can affect crops, and many serve as natural indicators and pollinators. In addition, a few of these species play the role of primary, secondary consumers or decomposers [6]. Organisms found in the Chordate phylum have high diversities of ecological niches, with notable terrestrial and aquatic environment adaptations [7]. In terms of ecosystemic level, the introduction of mammals can affect food chain functioning, generating a cascade effect on the composition and abundance of invasive predator, herbivore, and plant species as well as nutrient cycles [7].

Panama has the highest number of known vertebrate animals of any country in Central America or the Caribbean and more bird species than the United States and Canada together [8]. In addition, it has 3.5% of flowering plants and 7.3% of fern and related species in the world [9]. Panama is the twenty-eighth country in the world with the greatest biological diversity [8]. However, in proportion to its size, it ranks tenth [8]. 

Panama’s location and geography have facilitated the introduction of alien species [9]. Its geographical position, high land elevation, and maritime connectivity, as well as the existence of the Panama Canal, has helped biological invasions in Panama [9]. According to [10], the new locks used in the expansion of the Panama Canal allow the passage of ships that are 366 m longer and 49 m wider, facilitating higher transportation of exotic species through either the ship ballast or attachment to the ship’s hull [9]. 

According to the IV Panama National Biodiversity Report, an estimate of approximately 324 exotic species have been introduced into Panama, the majority of which are plants. In addition, the introduction of exotic pet species has increased. Among the exotic species used as pets are birds, reptiles, and mammals [9]; however, it is not known how many of these exotic species have become invasive species. Due to the lack of records of invasive exotic species in Panama, this study is necessary to help answer how many invasive species of the group of chordates and arthropods have been registered in Panama.

This study provides the first list of invasive species in the Panamanian territory of two important groups of organisms: arthropods and chordates. This study also serves as preliminary data on the sources of introduction and provides information for future research and plans to prevent the negative impact of those species.

## 2. Materials and Methods

### 2.1. Study Area

Panama is geographically located at 7°12′07″ and 9°38′46″ north latitude and 77°09′24″ and 83°03′07″ west Neotropical Region longitude. Panama has a land area of 75,416.6 km^2^ and is divided into ten provinces. Panama is in the central part of the American continent, in the most eastern and south part of Central America. It is bordered by the Caribbean Sea in the north, the Pacific Ocean in the south, Colombia in the east, and Costa Rica in the west [8].

The economy of Panama is based on four activities, one of them being the logistics industry, which is fundamentally based on the movement of cargo from all over the world [11]. This movement that takes place through ports, airports, railways, and the Panama Canal is one of the main activities that facilitates the introduction of invasive species [12].

### 2.2. Study Selection

The study was carried out through an exhaustive information review of the following sources:—Global Invasive Species Database: http://www.iucngisd.org/gisd/species.php?sc=965, accessed on 20 January 2023; Invasive Species Specialist Group IUCN/SSC; Invasive Species Specialist Group (ISSG); and Invasive Species Compendium (CABI): https://www.cabi.org/isc/datasheet/108530#tolistOfSpecies (accessed on 20 January 2023)—from bibliographic databases such as Web of Science (WOS), Science Direct, Scielo, PubMed, Redalyc, Dimensions, and Google Scholar. Using the Publish or Perish software version 8, the URLs resulting from the searches were downloaded in csv format, which was worked in Excel to filter those records that contained a mixture of words (invasive species, Panama alien species, invasive species in Panama, invasive exotic species in Panama, Panama invasive species). Additionally, a thorough search was carried out in specialized journals on the subject, such as *BioInvasions Records* (Reabic); *Biological Invasions* (Springer); *Revista Bioinvasiones*; *Check List—The Journal of Biodiversity Data* (PENSOFT); as well as the journals published in Panama that are found on the ABC Platforms (SENACYT), such as *National Resources* and *SIBIUP* of the University of Panama.

The search was done using the keywords invasive species, exotic species, invasive species in Panama, invasive exotic species in Central America, invasive species of insects in Panama, invasive species in America, new records + Panamá, pest + Panama, introduced species + Panamá, and “alien species” and filtered by the words Panama, new records + Panama, pest + Panama, and introduced species + Panama.

Due to the limited literature published on the subject for Panama, the country’s specialists from the different groups under study were asked for a list of the species considered invasive in the country. We visited the following organizations and reviewed their bibliographies on the subject: Environment Ministry, the Aquatic Resources Authority of Panama, the Panama Canal Authority, the Herbarium of the Panama University, and the Directorate of Plant Health of the Ministry of Agricultural Development.

Other sources of information used were *Fifth National Report of Panama to the Convention on Biological Diversity* [13]; *Invasives in Mesoamerica and the Caribbean* [14]; *National Biodiversity Strategy and Action Plan 2018–2050* [15], and the Ministerio de Desarrollo Agropecuario database of invasive species 2019. 

The information obtained was organized in tables for easier understanding. A database of the species was created (in cases where the information was obtained) with the following variables: report date (year), province, altitude, place of origin. This allowed us to answer the research question and achieve one of the objectives. Also, to make easy the access to information on invasive species of the phyla Arthropoda and Chordate, another two tables were created, one including the taxonomic identification of the groups and the reference according to the article or database that was obtained for the species and the other containing limitations on the distribution of pets under official control in the Panama Republic.

## 3. Results

Through the collection of information, the results indicated that approximately 141 invasive exotic species of the Arthropoda and Chordata phyla have been reported in Panama. Of the 141 species, 50 species belonged to the Arthropoda phylum and 91 species belong to the Chordate phylum, as indicated in taxonomy Table 1. 

Of the 50 species belonging to the Arthropoda phylum, 37 species belonged to the Insecta class, three to the Arachnida class, nine to the Malacostraca class, and one to the Maxillopoda class. Of the 91 species of the Chordate phylum, 49 belonged to the Actinopterygii class, 20 to the Ascidiacea class, eight species to the Reptilia class, five to the bird class, and four to the Amphibia and Mammalia class.

In the Arthropoda Phylum, 18 species belonged to the Coleoptera order, being the most abundant of the groups in this phylum, followed by Decapoda with nine species; Hymenoptera with six species; Diptera and Hemiptera with five each; Acarida with two species; and one species each in the Thysanoptera, Lepidoptera, Blattodea, Mesostigmata, and Cephalobaenida orders.

In the Chordate phylum, 24 species belonged to the Perciforme order; 11 belonged to the Stolidobranchia order; seven species each belonged to the Squamata and Cypriniformes orders; six species each belonged to the Cyprinodontiformes and Aplousobranchia orders; four to the Anura order; three each to the Passeriformes, Characiformes, and Phlebobranchia orders; two each to the Rodentia, Carnivora, Salmoniformes, and Gobiiformes orders; and one species each to the Columbiformes, Pelecaniformes, Carangiformes, Acanthuriformes, Scorpaeniformes, Elopiformes, Clupeiformes, Syngnathiformes, Siluriformes, and Testudines orders.

The three most abundant families of the Arthropoda phylum were Curculionidae (14 spp.), Formicidae (4 spp.), and Tephritidae (3 spp.). For the Chordata phylum, the three most abundant families were Cichlidae (11 spp.); Styelidae (8 spp.); and Cyprinidae, Poeciliidae, and Centrarchidae (6 spp.).

We labeled one case, *Rachycentron canadum* (Linnaeus, 1766) [9,13], as “under scrutiny”, as the data located it in the Pacific zone of the Panama Republic, but it has never been seen in other Panamanian aquatic ecosystems. 

Based on the information obtained, we can say that the province with the highest record of invasive species introduced in the Panama Republic is the Panama province (111 records), followed by Colon (21 records), Chiriqui province (19 records), Veraguas (15 records), Cocle province (13 records), West Panama (12 records), Herrera (10 records), Bocas del Toro (7 records), Los Santos and Darien (10 records), and Kuna Yala (1 record) (Table 2 & Figure 1). 

The four continents with the most record of invasive species origins that have been reported in Panama were Asia (22 records), South America (21 records), Africa (15 records), and North America (14 records) (Table 2). The results indicate that the majority of registered invasive species in Panamá are global invasive species (Table 2).

Among the species reported as invasive for the Panama Republic, nine species are under official control, all belonging to the insect class, for causing damage to cucurbit crops, tomato, citrus, coffee, and species of the Arecaceae family (Table 3).

## 4. Discussion

Since its emergence and closure, Panama’s Isthmus has been a transit node in global invasion flows. Due to the closure of Panama’s Isthmus, natural events such as the great American biotic exchange [8] have developed, and later during Panamá’s colonial times, due to its geographical position, Panama’s Isthmus was used to transfer merchandise that promoted accidental or intentional species exchange. Later, with the construction of the Panama Canal Railway, the movement of all types of cargo from all over the world increased, making Panama one of the countries with the most species movement globally. Likewise, the presence for almost 100 years of the so-called Panama Canal Zone (area around the Panama Canal under the jurisdiction of the United States) led to the introduction of exotic species to the country without any type of control, many of which became invasive [64].

The results indicate that the main Panamanian ecosystem occupied by invasive exotic species is the aquatic ecosystem, with fish being the most dominant. Most of these species were introduced intentionally and, in many cases, are used as a food source in both farming systems and reservoirs. In terrestrial ecosystems, insects are the largest recorded group of invasive species. The invasive species of this group have become pests in agricultural crops or forest plantations, reducing crop yields and affecting wood commercialization (Table 3). Other species of insects affect ornamental plants, in which case, the damage has been less quantified. In this group, important vectors of human viral diseases (dengue, zika, and chikungunya) are also registered.

In the case of amphibians and reptiles, documentation of the ecological importance of invasive species is scarce. Most records are associated with urban areas, and the extent of their distribution depends greatly on anthropogenic activities [62,63,64,65]. For example, *Eleutherodactylus johnstonei* uses the vegetation associated with the herbaceous substrate as places to vocalize or perch [66]. *Eleutherodactylus planirostris* [62] is an introduced frog that has most likely moved to disturbed locations, such as forest edges and outside of residential home gardens, that represent its known habitat [65]. On the other hand, *Trachemys scripta elegans* is a species introduced for the pet trade [44], and it prefers calm waters with soft and muddy bottoms, aquatic vegetation, and suitable places for sunbathing. Within their native range, red-eared sliders occupy an ecological niche as predators and prey. They are resistant and, outside their native area, they occupy the same ecological niche with great adaptability [67]. *A. sagrei*, for its part, mainly occupies open areas, is highly territorial, and can ecologically displace native species. It is well adapted to finding food and avoiding predators in newly colonized habitats. Therefore, its capacity to displace other native species may be high [59]. Some species have only been reported, but no studies have been carried out on their current status and ecological impact. Some examples are *Rana catesbeiana* [44], *Sphaerodactylus argus* [68], and *Eleutherodactylus antillensis* [44]. Authors have suggested that some of these invasive amphibians and reptiles can be categorized as pests [44]; however, there is no evidence to support this, so we prefer to categorize them as “Species with potential to displace native species” or “Scarcely documented species”.

In the case of mammals, despite being from only a few species, the effect on public health is immense, as is the case of *Mus musculus* and *Rattus norvegicus*. For their part, the two species of the Herpestes genus are aggressive predators that cause imbalances in the local fauna.

According to our results from the Arthropoda and Chordate phyla, there are an estimated 141 invasive species for the Panama Republic. Compared with the information from other Iberoamerican countries, we found that for the countries that are part of the Andean Community (Colombia, Ecuador, Venezuela, Peru, and Bolivia), 227 invasive exotic species have been identified, mostly plants (92 species), insect pests (61 species), and vertebrates (30 species) [48]. In Costa Rica, 235 invasive species have been reported from all groups of organisms [69]. For the Dominican Republic, 192 species have been reported, of which 38 species were fish, 4 were amphibians, 8 were reptiles, 13 were birds, and 13 were mammals [70]. While in other latitudes, such as the Iberian Peninsula (Spain, Portugal, and Andora), 100 invasive species have been recorded [71].

When comparing the number of invasive species based on the surface area of each country (Andean Community and Dominican Republic) where the study was conducted, we found that the number is higher for the Panama Republic (141 species) compared to the Andean Community (166 species) and Dominican Republic (138 species). 

In the Andean Community, the area of Panama is 75,416.6 km^2^; the area of Colombia is 1142 million km^2^; the area of Ecuador is 256,370 km^2^; the area of Venezuela is 916,445 km^2^; the area of Peru is 1285 million km^2^; the area of Bolivia is 1099 million km^2^; and the area of the Dominican Republic is 48,442 km^2^. This indicates that the Panama Republic, even though it is a small country compared to those previously mentioned, has quite a considerable number of invasive species in the territory.

The economic activity of the Dominican Republic is based on beaches and tourism [72]; Colombia on the free market (exchange of goods and services between people and companies through monetary transactions) [73]; Ecuador and Venezuela on oil revenues [74,75]; Peru on the exploitation, processing, and export of natural resources, mainly mining, agriculture, and fishing [76]; and Bolivia on natural gas [77]. Since the Panama Republic is a country that bases its economy on logistics and transportation, commerce, and finance, we can assume that this is the reason why the constant influx of invasive species is so high compared to the others countries.

The results clearly indicate that the Panama Province is where the greatest number of exotic invasive species have been recorded. This is because the Panama Province is where the largest movement of merchandise operations takes place through the Panama Canal, the Panama Railway, and two of the main seaports.

Invasive exotic species are one of the main threats to countries’ economies and local diversity [1,2]. In Panama, in the case of species that cause economic activity damage, the state permanently monitors the containers of fresh products that enter through different ports and airports of the country [78]. The surveillance programs of the Ministry of Agricultural Development include a list of quarantine species that are monitored for detention in customs facilities. This surveillance is carried out by highly qualified personnel to diagnose species considered of quarantine interest that may affect different crops [78]. Once the invasive exotic species evades the detection line and is introduced into the country, the state develops constant sampling programs in the different provinces to restrict the local distribution of the species (Table 3). An example of this is the Fruit Fly Monitoring Program, which through maximum trapping, has prevented species such as *Anastrepha grandis* or *Ceratitis capitata* from spreading throughout the country. The Panamanian state also invests resources in the control and monitoring of exotic species that are vectors of viral diseases such as dengue, zika, and chikungunya. In recent times, these public health programs developed by the Ministry of Health have extended to the border with the Colombian Republic due mainly to the massive migratory phenomenon across the Colombian–Panamanian border in recent years.

In reference to invasive exotic species that affect natural ecosystems, the situation is different. Due to its variety of natural ecosystems, Panama offers an innumerable number of habitats that invasive exotic species can occupy. According to our results, Panama City is the main gateway for invasive exotic species to enter the country. Panama City is surrounded by urban forests with different states of conservation and management, which offers habitats that are highly degraded or contaminated by exotic species. Likewise, the Panama Province and the Colon Province, which is where the Panama Canal is located and which separate the Pacific Ocean from the Caribbean Sea by only 82 km, offer a variety of aquatic habitats for the species that come on ships from different parts of the world. To address this situation and lack of information, the Panamanian state, through the Ministry of the Environment, has established the *National Biodiversity Strategy and Action Plan 2018–2030*, which addresses the problem of invasive species in the country [15].

## 5. Conclusions

The Panama Republic, due to its geographical position and commercial activity, is an important point of global biological invasion. Panamanian aquatic ecosystems are those that have been the most occupied by invasive species. However, invasive species of the insect group quantitatively cause the most damage to humans and the environment.

When comparing the number of invasive species based on the surface area of each country in which this study was conducted, we found that the number was highest for the Panama Republic.

Due to the limitations of this work, it is possible that some species were not included due to a scarcity of information, or that they have not been considered invasive for some researchers because they may correspond to natural distribution patterns, like the case of the coyote.

## Figures and Tables

**Figure 1 biology-13-00571-f001:**
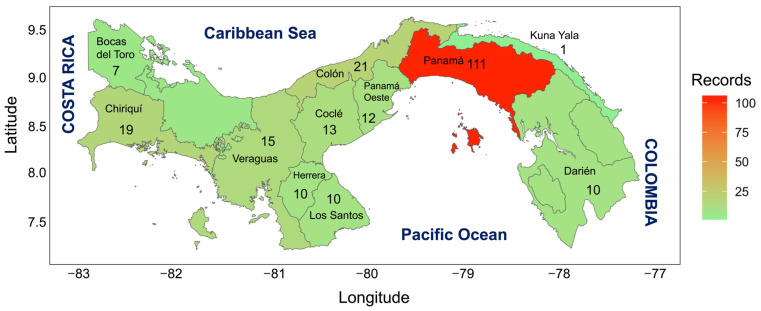
Map of the Republic of Panama indicating the number of invasive species by province.

**Table 1 biology-13-00571-t001:** A taxonomic list of invasive species recorded for the Panama Republic.

Species	Family	Order	Class	Ecological Status	Associated Organism or Habitat	Ref.
		ARTHROPODA				
*Hypothenemus hampei* Ferrari, 1867	Curculionidae	Coleoptera	Insecta	Pest species	Coffee	[16]
*Coccotrypes advena* Blandford 1894	Curculionidae	Coleoptera	Insecta	Pest species	Coffee	[17]
*Ambrosiodmus obliquus* LeConte, 1878	Curculionidae	Coleoptera	Insecta	Pathogen transmitter	Wood/forestry trees	[17]
*Coptoborus ricini* Eggers, 1932	Curculionidae	Coleoptera	Insecta	Pest species	Wood/forestry trees	[17]
*Xyleborus bispinatus* Eichhoff, 1864	Curculionidae	Coleoptera	Insecta	Fungal pathogen transmitter	Wood/forestry trees	[17]
*Xylosandrus morigerus* Blandford, 1894	Curculionidae	Coleoptera	Insecta	Pest species	Wood/forestry trees	[17]
*Xylosandrus compactus* Eichhinoff, 1876	Curculionidae	Coleoptera	Insecta	Pest species	Wood/forestry trees	[17]
*Cryptocarenus seriatus* Eggers, 1933	Curculionidae	Coleoptera	Insecta	Pest species	Wood	[17]
*Coccotrypes vulgaris* Eggers, 1923	Curculionidae	Coleoptera	Insecta	Pest species	Wood/trees	[17]
*Dendrocranulus tardus* Blandford, 1896	Curculionidae	Coleoptera	Insecta	Pest species	Wood/forestry trees	[17]
*Scolytopsis puncticollis* Blandford, 1896	Curculionidae	Coleoptera	Insecta	Pest species	Stem borer	[17]
*Xylosandrus crassiusculus* Motschulsky, 1866	Curculionidae	Coleoptera	Insecta	Pest species	Wood/forestry trees	[18]
*Xyleborinus exiguus* Walker, 1859	Curculionidae	Coleoptera	Insecta	Pest species	Tree polyphagous	[17]
*Rhynchophorus palmarum* Linneus, 1758	Curculionidae	Coleoptera	Insecta	Vector of the red ring (RR)	Palms	[19]
*Euoniticellus intermedius* Reiche, 1849	Scarabaeidae	Coleoptera	Insecta	Beneficial	Pasturelands	[20]
*Harmonia axyridis* Pallas, 1773	Coccinellidae	Coleoptera	Insecta	Pest species	Native species displace	[21]
*Callosobruchus phaseoli* Gyllenhal, 1833	Chrysomelidae	Coleoptera	Insecta	Pest species	*Cajanus cajan*	[22]
*Saperda candida* Fabricus, 1787	Cerambycidae	Coleoptera	Insecta	Pest species	Wood	[23]
*Brachyplatys subaeneus* Westwood, 1837	Plastaspidae	Hemiptera	Insecta	Pest species	*Cajanus cajan*	[24]
*Diaphorina citri* Kuwayama, 1908	Psyllidae	Hemiptera	Insecta	Pest species	Citrus plants	[25]
*Pseudacysta perseae* Heideman, 1908	Tingidae	Hemiptera	Insecta	Pest species	*Persea americana*	[26]
*Aulacaspis yasumatsui* Takagi, 1977	Diaspididae	Hemiptera	Insecta	Pest species	*Cycas* spp.	[27]
*Aphis spiraecola* Patch, 1914	Aphididae	Hemiptera	Insecta	Pest species	*Citrus*	[28]
*Thrips palmi* Karny, 1990	Thripidae	Thysanoptera	Insecta	Pest species	Cucurbitaceae Solanaceae	[29]
*Apis mellifera scutellata* Linneus, 1758	Apidae	Hymenoptera	Insecta	Pest species	Human	[30]
*Tapinoma melanocephalum* Fabricius, 1793	Formicidae	Hymenoptera	Insecta	Pest species	Human, native species displace	[23]
*Paratrechina longicornis* Motshulsky, 1863	Formicidae	Hymenoptera	Insecta	Pest species	Environmental damage	[23]
*Monomorium floricola* Jerdon, 1851	Formicidae	Hymenoptera	Insecta	Pest species	Native species displace	[23]
*Quadrastichus erythrinae* Kim, 2004	Eulophidae	Hymenoptera	Insecta	Pest species	*Erytrina* spp.	[31]
*Nylanderia fulva* Mayr, 1862	Formicidae	Hymenoptera	Insecta	Pest species	Crops	[32]
*Ceratitis capitata* Weidemann, 1824	Tephritidae	Diptera	Insecta	Pest species	Fruits	[33]
*Anastrepha obliqua* Macquart, 1835	Tephritidae	Diptera	Insecta	Pest species	Fruits	[34]
*Anastrepha grandis* Macquart, 1846	Tephritidae	Diptera	Insecta	Pest species	Cucurbitacea	[35]
*Aedes aegypti* Linnaeus, 1762	Culicidae	Diptera	Insecta	Disease vector	Human	[36]
*Aedes albopictus* Skuse, 1895	Culicidae	Diptera	Insecta	Disease vector	Human	[36]
*Tuta absoluta* Meyrick, 1917	Gelechiidae	Lepidoptera	Insecta	Pest species	Tomato and potato	[12]
*Blattella germanica* Linnaeus, 1767	Ectobiidae	Blattodea	Insecta	Pest species	Human	[37]
*Brevipalpus phoenicis* Geijskes, 1936	Tenuipalpidae	Acarida	Arachnida	Pest species	Citrus	[38]
*Brevipalpus californicus* Banks, 1904	Tenuipalpidae	Acarida	Arachnida	Pest species CiLV vector	Citrus	[39]
*Dermanyssus gallinae* De Geer, 1778	Dermanyssidae	Mesostigmata	Arachnida	Hematophagous ectoparasite	Human and other animals	[40]
*Rhithropanopeus harrisii* Gould, 1841	Panopeidae	Decapoda	Malacostraca	Pest species	Aquatic biome	[41]
*Elamenopsis kempi* Chopra y Das, 1930	Hymenosomatidae	Decapoda	Malacostraca	Pest species	Aquatic biome	[42]
*Procambarus clarkia* Girard, 1852	Cambaridae	Decapoda	Malacostraca	Pest species	Aquatic biome	[35]
*Penaeus monodon* Fabriciud, 1798	Penaeidae	Decapoda	Malacostraca	Pest species	Aquatic biome	[35]
*Cherax tenuimanus* Smith, 1912	Astacidae	Decapoda	Malacostraca	Pest species	Aquatic biome	[35]
*Macrobrachium amazonicum* Heller, 1862	Palaemonidae	Decapoda	Malacostraca	Pest species	Aquatic biome	[43]
*Macrobrachium rosenbergii* De Man, 1879	Palaemonidae	Decapoda	Malacostraca	Pest species	Aquatic biome	[44]
*Goniopsis cruentata* Latrelle, 1803	Grapsidae	Decapoda	Malacostraca	Pest species	Aquatic biome	[43]
*Pachygrapsyus gracilis* de Saussure, 1857	Grapsidae	Decapoda	Malacostraca	Pest species	Aquatic biome	[43]
*Raillietiella frenatus* Riley & Self, 1981	Cephalobaenidae	Cephalobaenida	Maxillopoda	Parasite	Native toads	[45]
		CHORDATA				
*Mus musculus* Linnaeus, 1758	Muridae	Rodentia	Mammalia	Pest species	Human	[23]
*Rattus norvegicus* Berkenhout, 1769	Muridae	Rodentia	Mammalia	Pest species	Human	[23]
*Herpestes auropunctatus* Hodgson, 1836	Herpestidae	Carnivora	Mammalia	Pest species	Health biodiversity	[23]
*Herpestes javanicus* Hilaire, 1818	Herpestidae	Carnivora	Mammalia	Pest species	Native species affect	[23]
*Columba livia* Gmelin, 1789	Columbidae	Columbiformes	Aves	Pest species	Human	[23]
*Bubulcus ibis* Linnaeus, 1758	Ardeidae	Pelecaniformes	Aves	Pest species	Human	[23]
*Mimus gilvus* Viellot, 1808	Mimidae	Passeriformes	Aves			[23]
*Passer domesticus* Linnaeus, 1758	Passeridae	Passeriformes	Aves	Pest species	Human	[46]
*Sicalis flaveola* Linnaeus, 1766	Thraupidae	Passeriformes	Aves	Pest species		[46]
*Rachycentron canadum* Linnaeus, 1766	Rachycentridae	Carangiformes	Actinopterygii	Incidental record	Marine biome	[47]
*Oreochromis niloticus* Linnaeus, 1758	Cichlidae	Perciformes	Actinopterygii	Pest species	Aquatic biome	[48,49]
*Oreochromis aureus* Steindachner, 1864	Cichlidae	Perciformes	Actinopterygii	Pest species	Aquatic biome	[50]
*Oreochromis mossambicus* Peters, 1852	Cichlidae	Perciformes	Actinopterygii	Pest species	Aquatic biome	[50]
*Oreochromis urolepis* Norman, 1912	Cichlidae	Perciformes	Actinopterygii	Pest species	Aquatic biome	[50]
*Coptodon rendalli* Boulenger, 1897	Cichlidae	Perciformes	Actinopterygii	Pest species	Aquatic biome	[50]
*Cichla ocellaris* Bloch & Schneider, 1801	Cichlidae	Perciformes	Actinopterygii	Pest species	Aquatic biome	[33]
*Cichla monoculus* Agassiz, 1831	Cichlidae	Perciformes	Actinopterygii	Pest species	Aquatic biome	[49,50]
*Parachromis managuensis* Günther, 1867	Cichlidae	Perciformes	Actinopterygii	Pest species	Aquatic biome	[49,51]
*Astronotus ocellatus* Agassiz, 1881	Cichlidae	Perciformes	Actinopterygii	Pest species	Aquatic biome	[49]
*Mesonauta festivus* Heckel, 1840	Cichlidae	Perciformes	Actinopterygii	Pest species	Aquatic biome	[49]
*Vieja maculicauda* Regan, 1905	Cichlidae	Perciformes	Actinopterygii	Pest species	Aquatic biome	[43]
*Eleotris melanosoma* Bleeker, 1853	Eleotridae	Gobiiformes	Actinopterygii	Pest species	Aquatic biome	[52]
*Butis koilomatodon* Bleeker, 1849	Butidae	Gobiiformes	Actinopterygii	Pest species	Aquatic biome	[52]
*Lepomis macrochirus* Rafinesque, 1819	Centrarchidae	Perciformes	Actinopterygii	Pest species	Aquatic biome	[44]
*Lepomis microlophus* Günther, 1859	Centrarchidae	Perciformes	Actinopterygii	Pest species	Aquatic biome	[53]
*Micropterus salmoides* Lacepede, 1802	Centrarchidae	Perciformes	Actinopterygii	Pest species	Aquatic biome	[49,53]
*Pomoxis annularis* Rafinesque, 1818	Centrarchidae	Perciformes	Actinopterygii	Pest species	Aquatic biome	[52]
*Pomoxis nigromaculatus* Lesueur, 1829	Centrarchidae	Perciformes	Actinopterygii	Pest species	Aquatic biome	[49,52]
*Lepomis humilis* Girard, 1858	Centrarchidae	Perciformes	Actinopterygii	Pest species	Aquatic biome	[49]
*Dormitator maculatus* Meek and Hildebrand, 1923	Eleotridae	Perciformes	Actinopterygii	Pest species	Aquatic biome	[43]
*Hypleurochilus pseudoaequipinnis* Bath, 1994	Blenniidae	Perciformes	Actinopterygii	Pest species	Aquatic biome	[43]
*Leptophilypnus fluviatilis* Meek and Hildebrand, 1916	Eleotridae	Perciformes	Actinopterygii	Pest species	Aquatic biome	[43]
*Lophogobius cyprinoides* Pallas, 1770	Gobiidae	Perciformes	Actinopterygii	Pest species	Aquatic biome	[43]
*Lupinoblennius vinctus* Poey, 1867	Blenniidae	Perciformes	Actinopterygii	Pest species	Aquatic biome	[43]
*Omobranchus punctatus* Valenncienes, 1836	Blenniidae	Perciformes	Actinopterygii	Pest species	Aquatic biome	[52]
*Sciaenops ocellatus* Linnaeus, 1766	Sciaenidae	Acanthuriformes	Actinopterygii	Pest species	Aquatic biome	[35]
*Oncorhynchus mykiss* Walbaum, 1792	Salmonidae	Salmoniformes	Actinopterygii	Pest species	Aquatic biome	[23,49]
*Salmo trutta* Linnaeus, 1758	Salmonidae	Salmoniformes	Actinopterygii	Pest species	Aquatic biome	[53]
*Pterois volitans* Linnaeus, 1758	Scorpaenidae	Scorpaeniformes	Actinopterygii	Pest species	Aquatic biome	[54]
*Cirrhinus molitorella* Valenciennes, 1844	Cyprinidae	Cypriniformes	Actinopterygii	Pest species	Aquatic biome	[35]
*Ctenopharyngodon Idella* Valenciennes, 1844	Cyprinidae	Cypriniformes	Actinopterygii	Pest species	Aquatic biome	[52]
*Cyprinus carpio* Linnaeus, 1758	Cyprinidae	Cypriniformes	Actinopterygii	Pest species	Aquatic biome	[23,49]
*Hypophthalmichthys molitrix* Valenciennes, 1844	Cyprinidae	Cypriniformes	Actinopterygii	Pest species	Aquatic biome	[52]
*Hypophthalmichthys nobilis* Richardson, 1845	Cyprinidae	Cypriniformes	Actinopterygii	Pest species	Aquatic biome	[23]
*Mylopharyngodon piceus* Richardson, 1846	Cyprinidae	Cypriniformes	Actinopterygii	Pest species	Aquatic biome	[35]
*Ictiobus cyprinellus* Valenciennes, 1844	Catostomidae	Cypriniformes	Actinopterygii	Pest species	Aquatic biome	[35]
*Gambusia holbrooki* Girard, 1859	Poeciliidae	Cyprinodontiformes	Actinopterygii	Pest species	Aquatic biome	[49]
*Xiphophorus helleri* Heckel, 1848	Poeciliidae	Cyprinodontiformes	Actinopterygii	Pest species	Aquatic biome	[49,55]
*Gambusia affinis* Baird & Girard, 1853	Poeciliidae	Cyprinodontiformes	Actinopterygii	Pest species	Aquatic biome	[55]
*Gambusia nicaraguensis* Günther, 1866	Poeciliidae	Cyprinodontiformes	Actinopterygii	Pest species	Aquatic biome	[55]
*Poecilia reticulata* Peters, 1859	Poeciliidae	Cyprinodontiformes	Actinopterygii	Pest species	Aquatic biome	[55]
*Xiphophorus variatus* Meek, 1904	Poeciliidae	Cyprinodontiformes	Actinopterygii	Pest species	Aquatic biome	[55]
*Ictalurus punctatus* Rafinesque, 1818	Ictaluridae	Siluriformes	Actinopterygii	Pest species	Aquatic biome	[35]
*Colossoma macropomum* Cuvier, 1816	Serrasalmidae	Characiformes	Actinopterygii	Pest species	Aquatic biome	[49,52]
*Piaractus brachypomus* Cuvier, 1818	Serrasalmidae	Characiformes	Actinopterygii	Pest species	Aquatic biome	[35]
*Piaractus mesopotamicus* Holmberg, 1887	Serrasalmidae	Characiformes	Actinopterygii	Pest species	Aquatic biome	[35]
*Microphis brachyurus lineatus* Kaup, 1856	Syngnathidae	Syngnathiformes	Actinopterygii	Pest species	Aquatic biome	[43]
*Megalops atlanticus* Valenciennes, 1847	Megalopidae	Elopiformes	Actinopterygii	Pest species	Aquatic biome	[43]
*Anchoa parva* Meek & Hildebrand, 1923	Engraulidae	Clupeiformes	Actinopterygii	Pest species	Aquatic biome	[43]
*Hemidactylus brookii* Gray, 1845	Gekkonidae	Squamata	Reptilia	Species with potential to displace native species	Terrestrial biome	[56]
*Hemidactylus frenatus* Dúmeril & Bibron, 1836	Gekkonidae	Squamata	Reptilia	Species with potential to displace native species	Terrestrial biome	[23,56]
*Hemidactylus mabouia* Moreau de Jonnes, 1818	Gekkonidae	Squamata	Reptilia	Scarcely documented species	Terrestrial biome	[23,56]
*Hemidactylus turcicus* Linnaeus, 1758	Gekkonidae	Squamata	Reptilia	Scarcely documented species	Terrestrial biome	[56,57]
*Lepidodactylus lugubris* Dúmeril & Bibron, 1836	Gekkonidae	Squamata	Reptilia	Species with potential to displace native species	Terrestrial biome	[56,58]
*Anolis sagrei* Cocteau in Dúmeril & Bibron, 1837	Polychrotidae	Squamata	Reptilia	Species with potential to displace native species	Terrestrial biome	[59]
*Sphaerodactylus argus* Gosse, 1850	Sphaerodactylidae	Squamata	Reptilia	Scarcely documented species	Terrestrial biome	[44,56]
*Trachemys scripta elegans* Wied, 1838	Emydidae	Testudines	Reptilia	Species with potential to displace native species	Semiaquatic	[23]
*Eleutherodactylus antillensis* Reinhardt & Lütken, 1863	Eleutherodactylidae	Anura	Amphibia	Scarcely documented species	Terrestrial biome	[60,61]
*Eleutherodactylus johnstonei* Barbour, 1914	Eleutherodactylidae	Anura	Amphibia	Scarcely documented species	Terrestrial biome	[60]
*Eleutherodactylus planirostris* Cope, 1862	Eleutherodactylidae	Anura	Amphibia	Scarcely documented species	Terrestrial biome	[62]
*Rana catesbeiana* Shaw, 1802	Ranidae	Anura	Amphibia	Scarcely documented species	Aquatic and semi-aquatic environments	[35]
*Ascidia incrassata* Heller, 1878	Ascidiidae	Phlebobranchia	Ascidiacea	Pest species	Aquatic biome	[63]
*Ascidia sydneiensis* Stimpson, 1855	Ascidiidae	Phlebobranchia	Ascidiacea	Pest species	Aquatic biome	[63]
*Phallusia nigra* Savigny, 1816	Ascidiidae	Phlebobranchia	Ascidiacea	Pest species	Aquatic biome	[63]
*Botrylloides nigrum* Herdman, 1886	Styelidae	Stolidobranchia	Ascidiacea	Pest species	Aquatic biome	[63]
*Botryllus planus* Van Name, 1902	Styelidae	Stolidobranchia	Ascidiacea	Pest species	Aquatic biome	[63]
*Polyandrocarpa anguinea* Sluiter, 1898	Styelidae	Stolidobranchia	Ascidiacea	Pest species	Aquatic biome	[63]
*Polyandrocarpa sagamiensis* Tokioka, 1953	Styelidae	Stolidobranchia	Ascidiacea	Pest species	Aquatic biome	[63]
*Polyandrocarpa zorritensis* Van Name, 1931	Styelidae	Stolidobranchia	Ascidiacea	Pest species	Aquatic biome	[63]
*Styela canopus* Savigny, 1816	Styelidae	Stolidobranchia	Ascidiacea	Pest species	Aquatic biome	[63]
*Symplegma brakenhielmi* Michaelsen, 1904	Styelidae	Stolidobranchia	Ascidiacea	Pest species	Aquatic biome	[63]
*Symplegma rubra* Monniot, 1972	Styelidae	Stolidobranchia	Ascidiacea	Pest species	Aquatic biome	[63]
*Herdmania pallida* Heller, 1878	Pyuridae	Stolidobranchia	Ascidiacea	Pest species	Aquatic biome	[63]
*Microcosmus exasperatus* Heller, 1878	Pyuridae	Stolidobranchia	Ascidiacea	Pest species	Aquatic biome	[63]
*Pyura vittata* Stimpson, 1852	Pyuridae	Stolidobranchia	Ascidiacea	Pest species	Aquatic biome	[63]
*Didemnum perlucidum* Monniot F, 1983	Didemnidae	Aplousobranchia	Ascidiacea	Pest species	Aquatic biome	[63]
*Didemnum psammatodes* Sluiter, 1895	Didemnidae	Aplousobranchia	Ascidiacea	Pest species	Aquatic biome	[63]
*Diplosoma listerianum* Milne Edwards, 1841	Didemnidae	Aplousobranchia	Ascidiacea	Pest species	Aquatic biome	[63]
*Lissoclinum fragile* Van Name, 1902	Didemnidae	Aplousobranchia	Ascidiacea	Pest species	Aquatic biome	[63]
*Trididemnum orbiculatum* Van Name, 1902	Didemnidae	Aplousobranchia	Ascidiacea	Pest species	Aquatic biome	[63]
*Polyclinum constellatum* Savigny, 1816	Polyclinidae	Aplousobranchia	Ascidiacea	Pest species	Aquatic biome	[63]

**Table 2 biology-13-00571-t002:** Bionomic data about the invasive species recorded for the Panama Republic.

Species	Province	District	Site	Introduction Year	Continent of Origin
*Hypothenemus hampei* ^1^	Chiriqui	Chiriqui	Border with Costa Rica	2007	Africa
*Coccotrypes advena* ^1^	Panama, Colon	Panama, Colón	Panama Canal	2007	Asia
*Ambrosiodmus obliquus* ^2^	Panama				
*Coptoborus ricini* ^2^	Panama				
*Xyleborus bispinatus* ^2^	Panama				
*Xylosandrus morigerus* ^1^	Panama				
*Xylosandrus compactus* ^1^	Panama				
*Cryptocarenus seriatus* ^2^	Panama				
*Coccotrypes vulgaris* ^2^	Panama				
*Dendrocranulus tardus* ^2^	Panama				
*Scolytopsis puncticollis* ^1^	Panama				
*Xylosandrus crassiusculus* ^1^	Panama	Gatun	Barro Colorado	2004	Asia
*Xyleborinus exiguus* ^1^	Panama	Gatun	Barro Colorado	2019	Asia
*Rhynchophorus palmarum* ^2^	Panama	Panama, Colon	Panama		
*Euoniticellus intermedius* ^1^	Chiriqui	Renacimiento	Chiriqui	2015	Africa
*Harmonia axyridis* ^1^	Colon	Colon	Centro Regional Colon	2014	Europe
*Callosobruchus phaseoli* ^2^	Panama	Arraijan	Las villas de Arraijan	2022	Neotropics
*Saperda candida* ^2^					
*Brachyplatys subaeneus* ^1^	Herrera	Chitre	Costa Pacífica	2016	Asia
*Diaphorina citri* ^1^	Bocas del Toro	Bocas del Toro	Bocas del Toro	2016	Asia
*Pseudacysta perseae* ^1^	Los Santos	Santa Maria	Peninsula de Azuero	2017	Europe
*Aulacaspis yasumatsui* ^1^	Panama	Panama	Balboa	2011–2021	Asia
*Aphis spiraecola* ^2^	Cocle	Penonome, Ola	Cocle	2002	South America
*Thrips palmi* ^1^	Herrera	Chitre	Los Chicharrones	2006	Asia
*Apis mellifera scutellata* ^1^	Panama	Panama	Panama Republic		Africa
*Tapinoma melanocephalum* ^1^					
*Paratrechina longicornis* ^1^					
*Monomorium floricola* ^1^					
*Quadrastichus erythrinae* ^1^	Panama	Panama	Panama Canal	2018	Asia
*Nylanderia fulva* ^2^	Chiriqui	Chiriqui	Chiriqui		
*Ceratitis capitata* ^1^	Chiriqui	Chiriqui	Chiriqui	1963	South America
*Anastrepha obliqua* ^2^	Cocle	Cocle	Cocle	2011	
*Anastrepha grandis* ^2^	Darien	La Palma	Darien	2009	South America
*Aedes aegypti* ^1^	Panama	Panama	Panama	1985	Africa
*Aedes albopictus* ^1^	Panama	Panama	Panama	2003	Asia
*Tuta absoluta* ^1^	Panama	Panama	Panama		
*Blattella germanica* ^1^	Panama	Panama	Panama		Africa
*Brevipalpus phoenicis* ^1^					
*Brevipalpus californicus* ^1^					
*Dermanyssus gallinae* ^1^	Herrera	Ocu	Las Guabas	2018	
*Rhithropanopeus harrisii* ^1^	Panama, Colon		Gatun Lake	1969	North America
*Elamenopsis kempi* ^1^	Panama, Colon		Gatun Lake	2007	Africa
*Procambarus clarkii* ^1^	Panama	Panama		1987	
*Penaeus monodon* ^1^	Panama	Panama	Panama		North America
*Cherax tenuimanus* ^1^	Panama	Panama	Panama		Oceania
*Macrobrachium amazonicum* ^2^	Panama, Colon		Gatun Lake		
*Macrobrachium rosenbergii* ^1^	Panama			1981	Asia
*Goniopsis cruentata* ^1^	Panama, Colon		Gatun Lake		
*Pachygrapsyus gracilis* ^1^	Panama, Colon		Gatun Lake		
*Raillietiella frenatus* ^1^	Panama				
*Mus musculus* ^1^	Panama	Panama	Panama	1973	Europe, Asia
*Rattus norvegicus* ^1^	Panama	Panama	Panama		Asia
*Herpestes auropunctatus* ^1^	Panama	Panama	Panama		Asia
*Herpestes javanicus* ^1^					
*Columba livia* ^1^	Panama	Panama	Panama	1608	Europe, Africa, Asia
*Bubulcus ibis* ^1^	Panama				
*Mimus gilvus* ^2^	Panama				
*Passer domesticus* ^1^	Panama	Panama	Panama	2019	Europe
*Sicalis flaveola* ^1^	Panama	Panama	Panama		South America
*Rachycentron canadum* ^1^	Panama		Gulf of Panama	2015	South America
*Oreochromis niloticus* ^1^	Chiriqui, Veraguas, Panama, Colon		Fortuna, La Yeguada, Gatun, Alajuela, Bayano and rivers	1976	Africa
*Oreochromis aureus* ^1^	Panama	Panama	Panama Canal	1976	Northwest Africa
*Oreochromis mossambicus* ^1^	Panama	Gatún, Alajuela	Panama Canal	1940	Southeast Africa
*Oreochromis urolepis* ^1^	Panama	Panama	Panama Canal	1979	Southeast Africa
*Coptodon rendalli* ^1^	Panama	Panama	Panama Canal	1977	Africa, Middle East
*Cichla ocellaris* ^1^	Panama, Colon		Gatun Lake	1967	South America
*Cichla monoculus* ^2^	Panama, Colon, Veraguas		Gatun, Alajuela y La Yeguada	1969	South America
*Parachromis managuensis* ^1^	Chirqui, Veraguas, Panama, Colon		Fortuna, La Yeguada, Gatun, Alajuela and rivers	1973	Central America (Atlantic Slope)
*Astronotus ocellatus* ^1^	Panama, Colon		Gatun Lake, Alajuela	1991	South America
*Mesonauta festivus* ^2^	Panama, Colon		Gatun Lake	1998–1999	South America
					Africa, Asia, Indo-Pacific
*Vieja maculicauda* ^2^	Chiriqui, Veraguas, Los santos, Herrera, Cocle, Panama Oeste, Panama, Darien			1939	
*Eleotris melanosoma* ^1^	Panama	Panama	Panama		
*Butis koilomatodon* ^1^	Panama	Panama	Miraflores Locks	1973	Occidental Indo-Pacific
*Lepomis macrochirus* ^1^	Chiriqui, West Panama, Panama		Lagunas de Volcan y San Carlos, Las Cumbres	1955	North America
*Lepomis microlophus* ^2^	Panama	Panama	Panama		North America
*Micropterus salmoides* ^1^	Chiriqui		Lagunas de Volcan	1955	North America
*Pomoxis annularis* ^2^	Panama	Panama	Panama		North America
*Pomoxis nigromaculatus* ^2^	West Panama	La Laguna	Laguna de San Carlos	1925–1935	North America
*Lepomis humilis* ^2^	West Panama	La Laguna	Laguna de San Carlos	1925–1935	North America
*Dormitator maculatus* ^2^	Panama	Panama	Panama Canal	1937	
*Hypleurochilus pseudoaequipinnis* ^2^	Panama	Panama	Panama Canal	1971	
*Leptophilypnus fluviatilis* ^2^	Panama	Panama	Panama Canal	1937	
*Lophogobius cyprinoides* ^2^	Panama	Panama	Panama Canal	1937	
*Lupinoblennius vinctus* ^2^	Panama	Panama	Panama Canal	1967	
*Omobranchus punctatus* ^1^	Panama	Panama	Panama Canal	2011	East Asia
*Sciaenops ocellatus* ^2^	Cocle	Aguadulce	El Salado	1987	North America
*Oncorhynchus mykiss* ^1^	Chiriqui		Chiriqui Viejo, Caldera, Boquete	1925	North America
*Salmo trutta* ^1^	Panama	Panama	Panama		Europe
*Pterois volitans* ^1^	Colon	Colon	Panamanian caribbean	2008	Pacific Ocean, Asia
*Butis koilomatodon* ^1^	Panama	Panama	Miraflores locks	1973	Occidental Indo-Pacific
*Cirrhinus molitorella* ^1^	Panama	Panama	Panama	1980	Asia
*Ctenopharyngodon idella* ^1^	Panama	Panama	Panama	1978	Asia
*Cyprinus carpio* ^1^	Chiriqui, Veraguas		Fortuna, La Yeguada	1976	Europe, Asia
*Hypophthalmichthys molitrix* ^1^	Panama	Panama	Panama	1978	Asia
*Hypophthalmichthys nobilis* ^1^	Panama	Panama	Panama	1978	Asia
*Mylopharyngodon piceus* ^1^	Panama	Panama	Panama		East Asia
*Ictiobus cyprinellus* ^2^	Veraguas	Santiago	Santiago	1987	Cuba
*Gambusia holbrooki* ^1^	Panama, Colon		Gatun Lake		North America
*Xiphophorus helleri* ^1^	Cocle, Panama		Las Cumbres, Anton Valley	1950’s y 1970’s	Central America (Atlantic Slope)
*Gambusia affinis* ^1^	Panama	Panama	Panama		
*Gambusia nicaraguensis* ^2^	Panama	Panama	Panama		
*Poecilia reticulata* ^1^	Veraguas, Panama		Santiago, Pacora, Chame, Las Cumbres, Cerro Azul	1910–1912	South America
*Xiphophorus variatus* ^2^	Panama	Panama	Panama		
*Ictalurus punctatus* ^1^	Panama	Panama	Panama	1981	North America
*Colossoma macropomum* ^2^	Panama, Chiriqui, Veraguas, Colon		La Yeguada, río Sereno y Alajuela	1980	South America
*Piaractus brachypomus* ^2^	Panama	Panama	Panama	1980	South America
*Piaractus mesopotamicus* ^2^	Panama	Panama	Panama	1986–1987	South America
*Microphis brachyurus lineatus* ^2^	Panama	Panama	Panama Canal	1971	Tropical region, Indo-Pacific
*Megalops atlanticus* ^2^	Panama	Panama	Panama Canal	1937	
*Anchoa parva* ^2^	Panama	Panama	Panama Canal	1937	
*Hemidactylus brookii* ^1^	Panama	Panama	Tocumen		
*Hemidactylus frenatus* ^1^	Panama	Panama	Chepo, Tocumen	2019	Tropical region, Indo-Pacific
*Hemidactylus mabouia* ^1^	Panama	Panama	Canal zone	1994	Sub-Saharan Africa
*Hemidactylus turcicus* ^1^	Panama	Panama	Canal Zone	1914	Mediterranean
*Lepidodactylus lugubris* ^1^	Not specific			2004	Sri Lanka, Africa
*Anolis sagrei* ^2^	Panama	Panama	Albrook	2018	Cuba
*Sphaerodactylus argus* ^2^	Bocas del Toro, San Blas	Bocas del Toro	Bocas del Toro, San Blas	2002	Jamaica, Caribbean Islands
*Trachemys scripta elegans* ^1^	Panama	Panama	Panama		United States
*Eleutherodactylus antillensis* ^1^	Panama	Panama	Panama City	1950–1960	Puerto Rico, Virgin Islands
*Eleutherodactylus johnstonei* ^2^	Panama	Panama	Panama	1980	
*Eleutherodactylus planirostris* ^2^	Panama	Panama	Panama		Cuba, Bahamas
*Rana catesbeianus* ^2^	Panama	Panama	Panama		North America
*Ascidia incrassata* ^1^	Panama	Panama	Panama Canal	2009	Indo-Pacific
*Ascidia sydneiensis* ^1^	Panama	Panama	Panama		Australia-Oceania
*Phallusia nigra* ^1^	Panama	Panama	Panama		Indo-Pacific
*Botrylloides nigrum* ^1^	Panama	Panama	Panama Canal	2002	South America
*Botryllus planus* ^2^	Panama	Panama	Panama		Indo-Pacific
*Polyandrocarpa anguinea* ^1^	Panama	Panama	Panama		Africa
*Polyandrocarpa sagamiensis* ^2^	Panama	Panama	Panama		South America
*Polyandrocarpa zorritensis* ^1^	Panama	Panama	Panama		South America
*Styela canopus* ^1^	Colon	Colon	Panama Canal		Indo-Pacific
*Symplegma brakenhielmi* ^1^	Panama	Panama	Panama		
*Symplegma rubra* ^1^	Panama	Panama	Taboguilla		Indo-Pacific
*Herdmania pallida* ^1^	Bocas del Toro	Colon Island	Bocas del Toro		South America
*Microcosmus exasperatus* ^1^	Panama	Panama	Panama		Colombia, South America
*Pyura vittata* ^1^	Panama	Panama	Panama		
*Didemnum perlucidum* ^1^	Panama	Panama	Panama		South America
*Didemnum psammatodes* ^1^	Panama	Panama	Panama		
*Diplosoma listerianum* ^1^	Panama	Panama	Panama		
*Lissoclinum fragile* ^1^	Panama	Panama	Panama		
*Trididemnum orbiculatum* ^2^	Panama	Panama	Panama		South America
*Polyclinum constellatum* ^1^	Panama	Panama	Panama		

Species with number: ^1^ = global invaders. ^2^ = local invaders [33].

**Table 3 biology-13-00571-t003:** The invasive species recorded for the Panama Republic for the first time (class Insecta).

Pests	Species	Host	Ecological Status
South American cucurbit fly	*Anastrepha grandis* Macquart, 1846	Cucurbitaceae	Pest species
Tomato moth	*Tuta absoluta* Meyrick, 1917	Tomato	Pest species
Huanglongbing (HLB)	*Diaphorina citri* Kuwayama, 1908	Citrus	Pest species
Mediterranean fly	*Ceratitis capitata* Wiedemann, 1824	Fruit plants (citrus, cucurbitaceae)	Pest species
Oriental *Trips palmi*	*Thrips palmi* Karny, 1990	Cucurbitaceae	Pest species
Coffee berry borer	*Hypothenemus hampei* Ferrari, 1867	Coffee	Pest species
Citrus leprosis virus	*Brevipalpus californicus* Banks, 1904	Citrus	Pest species
	*Brevipalpus phoenicis* Geijskes, 1936	Citrus	Pest species
Red ring disease	*Rhynchophorus palmarum* Linnaeus, 1758	Arecaceae	Pest species

## Data Availability

The data are included in the paper.

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
