# Peer review of "An Annotated Checklist of Invasive Species of the Phyla Arthropods and Chordates in Panama"

_biology, 2024, doi:10.3390/biology13080571_

Round 1
Reviewer 1 Report
Comments and Suggestions for Authors
The authors have done a lot of work to consolidate a large amount of information. The study area is a transit node in global invasion flows. The Isthmus of Panama has been a factor since the 15th century with the Panama Canal, after its opening in 1914, for the processes of invasion of aquatic organisms, and the Panama Canal is considered the center of global settlement. However, there is clearly insufficient information about invasive species in the canal region and Panama as a whole.
Thus, the presented research is relevant.
When analyzing the register of invasive species, information must be added:
1. the nature of the acquired habitat and the method of dispersal is “global”, for example Brevipalpus phoenicis, Passer domesticus, Polyclinum constellatum or “local” local Sicalis flaveola, type of organism – “free-living”, “parasite” – ectoparasites: Dermanyssus gallinae, Raillietiella frenatus Riley & Self, 1981, bloodsucking: Aedes aegypti, Aedes albopictus important in the spread of diseases (dengue, Chikungunya and Zika).
2. Particular attention should be paid to species of invaders of economic importance - agricultural pests, for example: Hypothenemus hampei - damaging coffee plantations, Anastrepha grandis - melons (pumpkins, zucchini, melons), Tuta absoluta - tomatoes, eggplant, potatoes and peppers, Brevipalpus phoenicis and Brevipalpus californicus - citrus fruits, tea, papaya, guava, coffee and other crops, as well as Xyleborus bispinatus, Xylosandrus morigerus, Xylosandrus compactus, Ambrosiodmus obliquus - damaging wood
3. Indicate the features of habitat and cultivation. Synanthropic organisms, insects: Blattella germanica and mammals: Mus musculus. Species active predators: mongooses: Herpestes auropunctatus and Herpestes javanicus, bullfrog Rana catesbeiana. Cultivated animals, for example in sea cages Rachycentron canadum. As well as domestic animals, geckos: Hemidactylus brookii, Hemidactylus frenatus, Hemidactylus mabouia Hemidactylus turcicus (https://doi.org/10.1093/cz/zox052)
4. Species occupying free niches, amphibians: Eleutherodactylus antillensis, so in Panama the species is found in urban areas, residential gardens and unoccupied lands (https://amphibiaweb.org/cgi/amphib_query?where-genus=Eleutherodactylus&where-species=antillensis&account= amphibiaweb).
5.Panama Canal factor in dispersal, for example: Elamenopsis kempi, Pachygrapsyus gracilis and ascidians.
6. Clarification of information:
- “Quiscalus mexicanus Gmelin, 1788 Icteridae Passeriformes Aves Chordata [55]” Inclusion in the list requires clarification, since Panama is within the limits of the natural and expansion of the natural range. At the same time, birds cause damage to crops in large flocks.
- “Trididemnum orbiculatum Van Name, 1902 Didemnidae Aplousobranchia Ascidiacea Chordata [73]” is a local invader, the country has a native and acquired part of the range “Two of these deal with the ascidian biota of Panama” (Carman et al. 2011, Bullard et al. . 2011).
General comments: it is necessary to improve the design, add maps of the study area, with a map of the number of invasive species by province. The information presented requires generalization and, in some cases, clarification. Indicate the provinces of the country with the highest number of types of invaders. Give more specific proposals for managing invasive processes in the country, taking into account the most significant consequences for agriculture and forestry, loss of biodiversity of aquatic and terrestrial ecosystems, and the spread of diseases.
Author Response
"Please see the attachment."

Reviewer 2 Report
Comments and Suggestions for Authors
I found this manuscript interesting and fairly well-written. However, there are a few issues with the language and writing style in some places; the Discussion and Conclusions also need some improvement. I recommend that the authors add the ecological status of the invasive species listed in the study. This addition will enhance the ecological value of the study. Once the authors make the suggested changes, the manuscript could be accepted for publication. Detailed comments are provided below.
Simple Summary
L22: “the numbers” instead of the “the amount”.
L28: I think “sources of income” here mean “sources of introduction”. Please fix this.
Abstract
LL31-32: what represents one the first barriers? This sentence seems to be missing something.
L33: “… , this study was necessary for identifying and registering ….” instead of the current sentence.
L35: “the search for invasive species …” instead of “the search of invasive species …”.
LL36-37: “The results show that …” instead of “The results indicate …”.
L40: “could facilitate the introduction of alien species into the country.” instead of the current.
L40: “This study provides …” instead of “The study …”.
Introduction
LL45-46: this statement needs a reference or a few references to support it.
L48: “, inducing changes that affect the native biodiversity.” instead of the current.
L51: what does “neither” refer too here. This sentence should be revised to improve clarity.
L54: “big economic losses” instead of the current.
L55” remove “in consequence”.
L56: “the need to include …” instead of the current.
L57: “keeping in mind” instead of “taking in mind”.
L57: I do not know what the authors mean by “nature dynamics” here. Please clarify.
L60: “challenges” instead of “barriers”.
L61: “since they are difficult to control and eradicate …” instead of the current.
LL63-65: this sentence should be integrated into the following paragraph.
L68: “bioindicators” instead of “natural indicators”.
L73: “food chain functioning” instead of “food chain function”.
LL59-75: this section is a bit choppy. I recommend that the authors combine these separate sections and paragraphs into a better-organized one paragraph.
L77: “has the highest number of …” instead of “has a higher number of …”.
L80: does “plants with flowers” here mean “flowering plants”? Please use accurate terminology.
LL80-82: the sentence is OK, but not very well-written. I would rewrite this sentence and clarify that Panama’s location and geography have facilitated the introduction of alien species.
LL82-85: also, not very well-written. I recommend something like: “facilitating higher transportation of exotic species through either ship ballast or attachment to ship hull.”
L87: “were introduced into Panama …” instead of the current.
L88” “exotic pet species” instead of “exotic species as pets”.
L91: “this study” instead of “this research”.
L93: “This study provides the first ….” Instead of the current.
L94: “This study also serves as …” instead of the current.
L95: “sources of introduction” instead of “sources of income” and “provides” instead of “will provide”.
L97: does “impact” here mean “the negative impact”?
Materials and Methods
LL102-104: “It is bordered by the Caribbean Sean in the north, the Pacific Ocean in the south, Colombia in the east, and Costa Rica in the west [12].” Instead of the current.
LL105-108: not very well-written. Better to rewrite into simpler and clearer language.
L109: “The economy of Panama is based on …” instead of the current.
L111: “takes places” instead of “develops”.
L112: “is one of the main activities that facilitates the introduction of invasive species.” instead of the current.
LL112-114: not very well-written. Better to rewrite into simpler and clearer language.
L151: “achieve on f the objectives” instead of “meet”.
L155: what is “pets limited distribution”? Do you mean the limited distribution of pets?
Results
L158: “The collected data show that approximately …” instead of the current.
Discussion
L222: “compared” instead of “comparing”.
L223: you need to adda comma before “we found”.
L228: “192 species have been reported, of which: 38 species were fish, 4 were amphibians, 8 were reptiles, 13 were birds, and 13 were mammals. [18].” Instead of the current.
L232: “where the study is conducted” instead of “addressed”.
L235: “The area of Panama is …” instead of the current.
LL241-249: not very well-written. Better to rewrite this section to improve quality and clarity.
You also need to discuss the reasons why Panama seems to have high numbers of invasive species. Some of those reasons were dealt with in the Introduction. It is better to discuss these possible causes in much detail in the discussion, and end the section with a proper concluding sentence.
Conclusions
L254: “The results show that …” instead of “indicate”.
L258: “conducted” instead of “addressed”.
L265: I am not sure what “dispersion” should mean here. Is it scarcity?
L267: also, I am not sure what dispersal patterns are supposed to mean here. Is it distribution patterns? Please clarify.
The conclusion is a bit repetitive. It could be integrated in the discussion after the causes of high numbers of invasive species in Panama as I suggested above.
Tables
Table 1: L186: “A taxonomy list of invasive species recorded for the Panama Republic” instead of the current title.
Table 2: “Bionomic data about the invasive species recorded for the Panama Republic” instead of the current title.
I also suggest that the authors add another column to specific the ecological status of the invasive species; is it a pest species, a beneficial species, or a pest species?
Table 3: “The invasive species recorded for the Panama Republic for the first time” instead of the current title.
I also suggest that the authors add another column to specific the ecological status of the invasive species; is it a pest species, a beneficial species, or a pest species?
Comments on the Quality of English LanguageComments on the language are included in the detailed report above.
Author Response
"Please see the attachment.

Round 2
Reviewer 1 Report
Comments and Suggestions for Authors
I thank the authors for the changes in the manuscript of the publication. It is also necessary to specify 1. species - global invaders and - local invaders; 2. indicate ways to reduce damage from invasive species: monitoring of diversity, compensatory measures and changes in the legislation of animal husbandry and aquaculture.
Author Response
"Please see the attachment."
